# Towards Interpretable Evaluations: A Case Study of Named Entity Recognition

## Abstract

With the proliferation of models for natural language processing (NLP) tasks, it is even harder to understand the differences between models and their relative merits. Simply looking at differences between holistic metrics such as accuracy, BLEU, or F1 do not tell us *why* or *how* a particular method is better and how dataset biases influence the choices of model design. In this paper, we present a general methodology for *interpretable* evaluation of NLP systems and choose the task of named entity recognition (NER) as a case study, which is a core task of identifying people, places, or organizations in text. The proposed evaluation method enables us to interpret the *model biases*, *dataset biases*, and how the *differences in the datasets* affect the design of the models, identifying the strengths and weaknesses of current approaches. By making our analysis tool available, we make it easy for future researchers to run similar analyses and drive the progress in this area.

## 1 Introduction

The development of deep neural networks has greatly sped the evolution of NLP systems. However, these advances have also come with a plethora of design decisions: should we choose a *CNN-based* (Kalchbrenner et al., 2014; Kim, 2014), *RNN-based* (Sutskever et al., 2014; Bahdanau et al., 2014) or *Transformer-based* (Vaswani et al., 2017; Dai et al., 2018) architecture? What variety of pre-training method should we use (Le & Mikolov, 2014; Peters et al., 2018; Devlin et al., 2018; Akbik et al., 2018)? The proliferation of model variants pose a great challenge for current evaluation methodology, which are usually opaque and simply give a single holistic score (Papineni et al., 2002; Banerjee & Lavie, 2005; Popović & Ney, 2011).

To alleviate this problem, researchers have made efforts, mainly focusing in two directions. First, some works (Farrús Cabeceran et al., 2010; Popović & Ney, 2011; Lommel et al., 2014) have attempted to shift the granularity of evaluation from holistic to fine-grained by conducting error analysis. Despite its effectiveness, the process of error analysis usually requires manual examination and depends on some pre-existing assumptions, suffering from confirmation bias, and risking ignoring new types of errors (Neubig et al., 2019). Additionally, this evaluation method based on error analysis is usually applied to only a single dataset (Karpathy et al., 2015; Kummerfeld & Klein, 2013; Kummerfeld et al., 2012), lacking discussion of fine-grained analysis in a *multi-dataset setting*. As a result, many important questions remain unclear: how to characterize the factors that influence the tasks for different datasets? how do the different choices of datasets influence the models' performance?

Another way to improve the evaluation strategy is common in our routine experimental design. That is to evaluate our models on multiple datasets with a holistic metric (Peters et al., 2018; Devlin et al., 2018). Currently, researchers are making efforts along this direction by setting up general evaluation benchmarks, such as GLUE (Wang et al., 2018) and SuperGLUE (Wang et al., 2019), which involve a diverse set of datasets. Although it enables us to gain a more comprehensive assessment of the models, the influence of different datasets on models is simply reflected by a holistic metric, which is not interpretable, and consequently, we are not clear about how different datasets influence the choices of model architectures.

In this paper, we argue that a complete evaluation method should not only reflect the individual performance of the model on one dataset or multiple datasets but also be able to *interpret* the model biases, dataset biases, and their correlation (how the difference in the datasets affects the design

of the models). We draw on the complementary strengths of the fine-grained evaluation and multi-dataset evaluation, driving fine-grained analysis to the multi-dataset setting. To this end, we devise a generalized evaluation methodology and choose the NER task as a case study. More specifically, we introduce the notion of *attribute*, which can be defined flexibly as the evaluation task needs. Here, we utilize the attribute to describe the property of each test entity for the NER task (i.e., entity length). Then, the test set will be divided into a set of *buckets* by different attributes of test entities. This makes it possible to evaluate recognition accuracy of different varieties of entities, achieving much more fine-grained analysis than standard corpus-level measures.

Additionally, the proposed attribute-aided evaluation methodology encourages us to introduce multiple attributes to find more potential factors which affect the NER models on different datasets. We further propose three analytical approaches as shown in Fig. 1: attribute-wise, model-wise, and bucket-wise analyses that have the following characteristics accordingly: **Attribute-wise** (Sec. 3.3.2) analysis could instruct us to find which factors matter for the NER tasks and figure out the commonality of factors across different NER datasets; **Model-wise** (Sec. 3.3.1) analysis aims to investigate how different attributes influence the performance of models with different architectures and pre-trained knowledge; **Bucket-wise** (Sec. 3.3.3) analysis diagnoses the strengths and weaknesses of existing models and helps us understand how different choices of datasets influence the model performance.

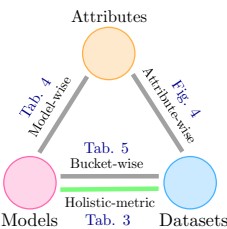

Figure 1: Relation chart among attributes, models, and datasets. The attribute-aimed method could bridge the gap between the model biases and dataset biases.

Our contributions can be precisely summarized as:

1) We draw on the complementary strengths of the fine-grained evaluation and multi-dataset evaluation, proposing a generalized evaluation methodology to interpret model biases, dataset biases, and their correlation. 2) We choose the NER task as a test case, the observations based on the extensive experiments (twelve models, ten attributes and six datasets) suggest directions for improvement and can drive the progress of this area. 3) Although some attributes defined in this paper are task-dependent, we claim our methodology is general since: a) for sequence labeling tasks (i.e., Part-of-Speech, text chunking, and extractive summarization), this evaluation method could be transferred without much modification. b) for other types of NLP tasks, we could re-define the attributes and our proposed bucketization strategies, as well as three analytic approaches, are task-agnostic. For example, in machine translation, the attribution could be sentence length (Luong et al., 2015), word itself (n-gram) in the reference file (Kumar & Tsvetkov, 2018); In question answering, the attribution could be answer length/type/position (Seo et al., 2016), document length (Joshi et al., 2017), and query length/type (Chen et al., 2016b). Once we have determined related attributes, we could make similar analyses based on our proposed measures.

## 2 PRELIMINARIES

We first summarize and compare past evaluation methodologies and then describe the NER task and its current evaluation strategy.

### 2.1 PROPERTIES OF EVALUATION METHODOLOGIES

Evaluation is gaining increasing interest in NLP, especially on text generation tasks as exemplified by machine translation (Fishel et al., 2012; Irvine et al., 2013; Daems et al., 2014). Generally, different evaluation methods can be characterized by the following three main properties:

*Interpretability*: Evaluation method could give interpretable results to help us understand where the weaknesses and strengths are. For example, "error analysis"(Kummerfeld et al., 2012;

| Methodology | Interp. | Conform. | Supple-exam |
|---|---|---|---|
| Holistic metric | × | × | × |
| Multi-dataset | × | × | ✓ |
| Error analysis | ✓ | ✓ | × |
| Diagnostic test | ✓ | ✓ | ✓ |
| Interpretable metric | ✓ | × | ✓ |

Table 1: Evaluation methodologies characterized by different properties.

Kummerfeld & Klein, 2013; Karpathy et al., 2015) is an interpretable evaluation method since we could figure out detailed limitations of the evaluated systems.

*Confirmation bias*: It represents a tendency to make tests consistent with the beforehand hypothesis. For example, Mudrakarta et al. (2018) assume that deep learning model are sensitive to question words in question answering tasks and verify it by carefully-designed adversarial examples, and Chen et al. (2016a) pre-defined six error to classify error cases.

*Supplementary exam*: It is an additional exam (require extra test sets ) for more comprehensive observations. i.e. multi-dataset evaluation (Devlin et al., 2018) on GLUE. Recently, there is a trend going from traditional evaluation to *diagnostic test*, in which a supplementary test set is required. For example, Jia & Liang (2017) propose an adversarial test for reading comprehension task and Naik et al. (2018) present a stress test method to diagnose natural language inference systems.

## 2.2 NER Task and Current Evaluation Strategy

**Task Description**    Named entity recognition (NER) is usually formulated as a sequence labeling problem (Huang et al., 2015; Ma & Hovy, 2016). Formally, let $X = \{x_1, x_2, \ldots, x_T\}$ be an input sequence and $Y = \{y_1, y_2, \ldots, y_T\}$ be the output tags. The goal of this task is to estimate the conditional probability: $P(Y|X) = P(y_t|X, y_1, \cdots, y_{t-1})$

**Evaluation Strategy for NER**    Existing NER systems are commonly evaluated by corpus-level metrics ($F1$-score) (Sang & De Meulder, 2003) and a small amount of work will conduct some manual error analysis (Ichihara et al., 2015; Derczynski et al., 2015). With the increasing improvement in network architectures and pre-trained knowledge, the NER systems are quickly reaching a performance plateau (Akbik et al., 2018; Akbik et al.). Therefore, fine-grained evaluation is required to identify the specific issues of existing NER systems. On the other hand, with the emergence of more and more NER datasets (Sang & De Meulder, 2003; Collobert et al., 2011; Weischedel et al., 2013), the time is ripe for us to bridge the gap between the insufficient understanding of the nature of datasets itself and model designs. In this paper, we choose the NER task as a test case, interpreting model biases, dataset biases, as well as their correlation under a general framework. This work also takes a step towards interpretable architecture searching.

## 3 Attribute-aided Evaluation Methodology

Our proposed evaluation methodology involves three key elements: *attribute definition* (Sec. 3.1), *bucketization* (Sec. 3.2) and the *analytical approach* (Sec. 3.3). Specifically, we first introduce the notion of *attribute* and define it in NER task as some property of the test entity, bywhich the test set will be divided into different sub-sets and the overall performance could be broken down into interpretable categories. Below, we will detail the three key elements.

### 3.1 Definition of Entity Attributes

*Entity Attributes* refer to the properties that can be used to characterize a given entity. Generally, different types of entity attributes provide different observation angles of system's performances.

Next, we will introduce entity attributes we explored in this paper in terms of *token level*, *span level* and *sentence level*. We take them into consideration since they are general features and could be transferred to other tasks.

**Token-level** 1) Token itself: it denotes the words of an entity. 2) Morphology: morphological features plays an essential role in many NLP tasks. Here, we define five cases for each entity token: including token, is upper case, lower case, digit, beginning with a capital letter, and others (such as punctuation).

**Span-level** 1) Entity itself: each entity is regarded as a unique identifier. 2) Entity length: the number of tokens in an entity. 3) Entity tag: the NER tag of an entity.

**Sentence-level** 1) Sentence length; 2) Entity density: it is the number of entity tokens divided by sentence length. 3) OOV density: it is the number of oov word divided by sentence length.

### 3.2 BUCKETIZATION STRATEGIES

*Bucketization* is an operation that breaks down the holistic performance into different interpretable categories based on the attribute values. This can be achieved by dividing the set of test entities into different subsets of test entities (for span- and sentence-level attributes) or test tokens (for token-level attributes). Without loss of generality, we describe the entity-based bucketization strategies while it can be easily applied to token-based. To legibly describe the studied problem, we follow the commonly used notations throughout the paper. We refer to $E$, $P$, $K$ as the sets of entities, entity attributes and attributes values, respectively.

The bucketization process can be formulated in a general form: $E_1^{te}, \cdots, E_m^{te} = \text{Bucket}(E^{te}|E^{tr}, p)$, where $E^{tr}$, $E^{te}$ represent the sets of training and test entities, respectively, $p$ denotes a type of the attribute. The basic idea is that the test entity set is divided into $m$ buckets based on the attribute $p$ and corresponding training set. where $E^{tr}$, $E^{te}$ represent the sets of training and test entities, respectively, $p$ denotes a type of the attribute.

Specifically, each subset of test entities can be obtained as follows: $E_i^{te} = \{e|\text{Atr}(e, p) \in \hat{K}_i, \forall e \in E^{te}\}$, where $\text{Atr}(e, p)$ is to query out the value of attribute $p$ for entity $e$. $\hat{K}_i$ is a set of attribute values, with which entities should be put into the $i$-th bucket: $\hat{K}_i = \{k|f(k) = i, \forall k \in K\}$. Above equation shows the key part for bucketization: *how do we build the relationship between the value of entity attributes and the bucket number?*. That is, we need to determine a criterion $f$, which guides us to put test entities according to its attribute values into suitable buckets. Here we explore three types of strategies for bucketization.

| Attributes | | Bucketization Strategies | | |
|---|---|---|---|---|
| | | R-Bucket | F-Bucket | MF-Bucket |
| Token | Token itself (F-tok) | | ✓ | |
| | Morphology (R-mor) | ✓ | | |
| Span | Entity itself (F-ent) | | ✓ | |
| | Entity len. (R-eLen) | ✓ | | |
| | Entity tag (F-tag) | | ✓ | |
| Sent | Sent length (R-sLen) | ✓ | | |
| | Entity dens. (R-eDen) | ✓ | | |
| | OOV dens. (R-oov) | ✓ | | |
| Multi | Entity & tag (MF-et) | | | ✓ |
| | Token & tag (MF-tt) | | | ✓ |

Table 2: Entity attributes we used in this paper and their corresponding bucketization strategies.

**Strategy-I: Range division of attribute values (*R-Bucket*)** An intuitive strategy is to bucketize the test entities based on their attribute values directly. Specifically, we could divide the range of attribute values into $m$ discrete parts. For example, the `sentence length` with range $[1, \cdots, 6]$ could be divided into $[1, 2]$, $[3, 4]$, $[5, 6]$ three buckets. Tab. 2 shows which attributes are matched to this bucketization strategy.

**Strategy-II: Familiarity of attribute values (*F-Bucket*)** The aim of the evaluation is to quantify the generalization errors of the system on the unseen samples. Therefore, by taking into account the degree to which the testing entities (or their attributes) have been seen in the training set, we can better figure out the impact of this attribute on model performance.

To achieve this, here we introduce a notion of *familiarity* to quantify the degree to which the attribute of a test entity has been seen in the training set.

$$F_k(p) = \frac{|\{e|\text{Atr}(e, p) = k, \forall e \in E^{tr}\}|}{|\{e \in E^{tr}\}|} \tag{1}$$

Here, $F_k(p) \in [0, 1]$ denotes the degree to which the test entity attribute $p$ with value $k$ have been seen in the training set. Then we can define the following criterion to achieve the bucketization: $F_{\hat{K}_i}(p) \in (\frac{i}{m}, \frac{i+1}{m}]$. The basic idea behind the criterion is that test entities could be put into the $i$-th bucket when the familiarity of their attribute values meets the above condition. Tab. 2 shows which attributes we used for *F-Bucket*.

**Strategy-III: Multi-attribute Familiarity (*MF-Bucket*)** The benefit of our general methodology for interpretable evaluation is that we can easily define new valuable measures based on old measures already defined. Here, we can adapt our *F-Bucket* strategies to multi-attribute version and modify the

Eq.1 as follows:

$$F_k(p_1, p_2) = \frac{|\{e|\mathrm{Atr}(e, p_1) = k_1 \wedge \mathrm{Atr}(e, p_2) = k_2, \forall e \in E^{tr}\}|}{|\{\mathrm{Atr}(e, p_1) = k_1, \forall e \in E^{tr}\}|} \tag{2}$$

For example, when we instantiate the two entity attributes $p_1$, $p_2$ as `entity itself` and `tag` respectively, the familiarity $F_k(p_1, p_2)$ is a measure with intriguing explanation: for each test entity with tag $k$, this measure quantifies its *category ambiguity*: the probability that this entity is labeled as $k$ in the training set.

To give a more intuitive understanding of the two processes: *attribute definition* and bucketization, Fig. 2 shows a concrete example, in which a set of attributes are defined for the target entity `New York`, with corresponding values.

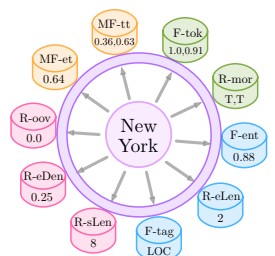

Figure 2: The attribute definition of the entity `New York` in the sentence: "No new fixtures reported from `New York`."

### 3.3 ANALYTIC APPROACHES

To better characterize the relation among attributes, models and datasets, we propose three analytical approaches: attribute-wise, model-wise, and bucket-wise, which can be used to interpret the model biases, dataset biases, and their correlation.

Formally, we refer to $M = m_1, \cdots, m_{|M|}$ as a set of **models** and $P = p_1, \cdots, p_{|P|}$ as a set of **attributes**. As describe above, the test set $E$ could be split into different **buckets** $E = E_1^j, \cdots, E_{|E|}^j$ based on a attribute $p_j$. We introduce the notion of performance table $\mathcal{T} \in \mathbb{R}^{|M| \times |P| \times |E|}$, in which $\mathcal{T}_{ijk}$ represents the performance of $i$-th model on the $k$-th sub-test set (bucket) generated by $j$-th attribute. Next, we will show how these approaches are defined based on $\mathcal{T}$.

#### 3.3.1 MODEL-WISE

The model-wise analysis aims to investigate how different attributes influence the performance of models with different architectures and pre-trained knowledge. For example, "does the `lengths of entities` influence the performances of `CNN-LSTM-CRF`-based NER system?"

Here we adopt two types of statistical variables $\mathbf{S}_{i,j}^\rho$ and $\mathbf{S}_{i,j}^\sigma$ to characterize how the $j$-th attribute influences the $i$-th model.

$$\mathbf{S}_{i,j}^\rho = \mathrm{Spearman}(\mathcal{T}[i, j :], R_j) \tag{3}$$
$$\mathbf{S}_{i,j}^\sigma = \mathrm{Std}(\mathcal{T}[i, j :]) \tag{4}$$

where Spearman is a function to calculate Spearman's rank correlation coefficient (Mukaka, 2012) and $R_j$ is the rank values of buckets based on $j$-th attribute. $\mathrm{Std}(\cdot)$ is the function to compute standard deviation.

Intuitively, $\mathbf{S}_{i,j}^\rho$ reflects the degree to which the $i$-th model positively (or negatively) correlates with $j$-th attribute while $\mathbf{S}_{i,j}^\sigma$ indicates the degree to which this attribute influences the model.

#### 3.3.2 ATTRIBUTE-WISE

The attribute-wise analysis aims to quantify the degree to which each attribute influences the NER task. To achieve this, we introduce four measures: task-independent variable $\zeta_j$ and task-dependent variables $\rho_j^1$ $\rho_j^2$ and $\sigma_j$ based on Eq.3 and Eq.4:

1) $\zeta_j = \frac{1}{|N|} \sum_i^{|N|} \mathrm{Atr}(e_i, j)$, where $N$ is the number test entities and $\mathrm{Atr}(e_i, j)$ represents the value of attribute $j$ for entity $e_i$.

For example, when $j$ represents the attribute of sentence length, $\zeta_j$ is the average sentence length of the whole dataset.

2) $\rho_j^1 = \frac{1}{|M|} \sum_i^{|M|} |\mathbf{S}_{i,j}^\rho|$, $\rho_j^2 = \frac{1}{|M|} \sum_i^{|M|} \mathbf{S}_{i,j}^\rho$, $\sigma_j = \frac{1}{|M|} \sum_i^{|M|} \mathbf{S}_{i,j}^\sigma$, where $|M|$ is the number of evaluated models. Compared with $\rho_j^1$, $\rho_j^2$ can reflect whether the correlation is positive or negative.

Intuitively, a higher absolute value of $\rho_j^1$ or $\rho_j^2$ suggests that attribute $j$ is a crucial factor, greatly influencing the performance of NER systems.

### 3.3.3 BUCKET-WISE

The bucket-wise analysis diagnoses the strengths and weaknesses of existing models. Moreover, based on attribute-wise analysis, we could understand how different choices of datasets influence the models' performance.

To this end, we introduce the following measures:

$$\beta_j = \begin{cases} \max_k(\mathcal{T}[a,j,k] - \mathcal{T}[b,j,k]) & (\mathcal{T}[a,j,k] > \mathcal{T}[b,j,k], \text{ for } \forall k) \\ \min_k(\mathcal{T}[a,j,k] - \mathcal{T}[b,j,k]) & \text{otherwise} \end{cases} \tag{5}$$

Usually where $a$, $b$ represent two different models and usually model $a$ has a higher performance (by dataset-level metric).

Intuitively, a negative value of $\beta_j$ suggests that a worse-ranked model (b) outperform the best-ranked model (a) in some aspect (attribute $j$); By contrast, a positive value shows the largest margin on the attribute $j$.

## 4 EXPERIMENTAL SETTINGS

### 4.1 MODELS AND ATTRIBUTES

**Model Settings** To evaluate the importance of different components of the NER systems, we varied our models mainly in terms of three aspects: different choices of character- (ELMo (Peters et al., 2018), Flair (Akbik et al., 2018; Akbik et al.)), subword- (BERT (Peters et al., 2018; Devlin et al., 2018)), word- (GloVe(Pennington et al., 2014)), and sentence-level encoders (LSTM (Hochreiter & Schmidhuber, 1997), CNN (Kalchbrenner et al., 2014)) and decoders (MLP or CRF (Lample et al., 2016; Collobert et al., 2011)). Detailed setting are shown in Tab.3. Totally, we study 12 NER models based on deep neural networks and one traditional method utilizing CRF Lafferty et al. (2001). The hyper-parameter settings of our evaluated models are shown in appendix section.

| Models | Character | | | | | Word | | | Sentence | | Decoder | | Overall F1 | | | | | |
|---|---|---|---|---|---|---|---|---|---|---|---|---|---|---|---|---|---|---|
| | none | cnn | elmo | flair | bert | none | rand | glove | lstm | cnn | crf | mlp | CoNLL | WNUT | BN | BC | MZ | WB |
| *CnonWrandlstmCrf* | ✓ | | | | | | ✓ | | ✓ | | ✓ | | 78.13 | 17.24 | 80.36 | 66.17 | 73.89 | 49.80 |
| *CcnnWnonelstmCrf* | | ✓ | | | | ✓ | | | ✓ | | ✓ | | 77.01 | 22.73 | 77.96 | 65.01 | 79.05 | 47.31 |
| *CcnnWrandlstmCrf* | | ✓ | | | | | ✓ | | ✓ | | ✓ | | 83.80 | 22.57 | 83.59 | 71.57 | 78.85 | 52.14 |
| *CcnnWglovelstmCrf* | | ✓ | | | | | | ✓ | ✓ | | ✓ | | 90.48 | 40.61 | 86.78 | 76.04 | 85.39 | 60.17 |
| *CcnnWglovecnnCrf* | | ✓ | | | | | | ✓ | | ✓ | ✓ | | 90.14 | 36.21 | 86.42 | 76.74 | **88.10** | 49.10 |
| *CcnnWglovelstmMlp* | | ✓ | | | | | | ✓ | ✓ | | | ✓ | 88.05 | 32.84 | 84.07 | 70.00 | 81.09 | 56.61 |
| *CelmWnonelstmCrf* | | | ✓ | | | ✓ | ✓ | | ✓ | | ✓ | | 91.64 | 44.56 | **89.75** | 77.10 | 86.32 | 60.51 |
| *CelmWglovelstmCrf* | | | ✓ | | | | | ✓ | ✓ | | ✓ | | 92.22 | 45.33 | 89.35 | 78.71 | 85.70 | 63.26 |
| *CbertWnonelstmMlp* | | | | | ✓ | ✓ | | | ✓ | | ✓ | | 91.11 | 42.50 | 89.64 | **81.03** | 86.90 | **66.35** |
| *CflairWnonelstmCrf* | | | | ✓ | | ✓ | | | ✓ | | ✓ | | 89.98 | 41.49 | 87.98 | 77.46 | 84.11 | 56.71 |
| *CflairWglovelstmCrf* | | | | ✓ | | | | ✓ | ✓ | | ✓ | | **93.03** | **45.96** | 87.92 | 77.23 | 85.56 | 63.38 |

Table 3: Neural NER systems with different architectures and pre-trained knowledge studied in this paper. Overall F1 shows the performances of corresponding systems on different datasets. For the model name, "C" refers to "Character" and "W" refers to "Word". Intuitively, the models are named based on their constituents. For example, *CnonWrandlstmCrf* is a model without character feature. Its word embedding is randomly initialized, and sentence encoder, as well as the decoder, are LSTM and CRF, respectively.

**Attributes** In our evaluation methodology, entity attributes can be defined flexibly and attribute values can be continuous or discrete. In this paper, although we investigate 10 types of attributes (or their combinations) as listed in Tab.2, others can be easily introduced.

## 4.2 NER DATASETS FOR EVALUATION

We conduct experiments on three benchmark datasets: the CoNLL2003 NER dataset, the WNUT16 dataset, and Ontonotes 5.0 dataset. The CoNLL2003 NER dataset (Sang & De Meulder, 2003) is based on Reuters data (Collobert et al., 2011). WNUT16 dataset is provided by the second shared task at WNUT-2016. The Ontonotes 5.0 dataset (Weischedel et al., 2013) is collected from newsgroups, broadcast news (BN), broadcast conversation (BC) and weblogs (WB) and magazine genre (MZ).

## 5 ANALYSIS

### 5.1 HOLISTIC ANALYSIS

Before giving a fine-grained analysis, we present the results of different models on different datasets in the way that traditional *multi-dataset* evaluation does. As shown in Fig. 3, we observe that there is no one-size-fits-all model, and the models with the best results on different datasets are different. Naturally, the following questions are raised: 1) What factors of the datasets can distinguish themselves and influence the NER task? 2) How do these factors influence the choices of models? 3) Does a worse-ranked model outperform the best-ranked model in some aspect and how the datasets influence the choices of models? The following analyses will be conducted around these questions.

### 5.2 ATTRIBUTE-WISE ANALYSIS

Attribute-wise measures enable us to characterize the dataset biases quantitatively. Here we utilize a radar chart in Fig. 4 to strikingly display the commonality and speciality between different datasets based on three measures $\zeta$, $\rho^1$, $\sigma$ defined in Sec. 3.3.2. And we also illustrate measure $\rho^2$ in Fig. 3. Detailed observations are listed as follows:

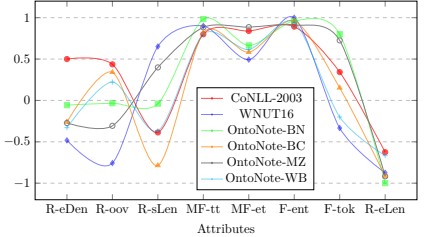

Figure 3: Illustration of task-dependent measure $\rho^2$.

*Category ambiguity* and *entity length* **have more consistent influence on NER performance.** The common parts of the radar chart Fig. 4 (b-c) illustrate that no matter which datasets are, the performance of NER task is highly correlated with these attributes: MF-tt (token-tag with MF-buck), MF-et (entity-tag with MF-buck), R-eLen (entity length with R-bucket). Fig. 3 shows the same result. This suggests that the prediction difficulty of named entity is commonly influenced by *category ambiguity* (MF-tt, MF-et), *entity length* (R-eLen).

*Occurence* and *sentence length* **matters but are minor factors.** The outliers in radar chart show the peculiarities of different datasets. Intuitively, on attributes: R-sLen, F-token, R-oov, the extent to which different datasets are affected varies greatly. Typically, as observed from Fig. 4-(a), a sorted sequence could be obtained according to the attribute F-tok: BN > MZ > WNUT > CoNLL > BC > WB. The reason why the Spearman correlations $\rho^1$ of BC and WB are smaller is that performance on test entities with higher-frequency tokens are lower than entities with lower-frequency tokens. This suggests that F-tok is not a decisive attribute and higher-frequency token can not guarantee a better performance since other crucial factors such as *category ambiguity* also matter.

*Entity density* **is a swing factor and CoNLL dataset is an outlier.** As shown in Fig. 3, the measure $\rho^2$ enables us to know the correlation is positive or negative. We observe that most datasets except CoNLL 2003 have negative Spearman values $\rho^2$ on the attribute of R-eDen, which suggests that a sentence with more entities is relatively harder to process. We can explain the unusual behavior on CoNLL 2003 from its intrinsic value $\zeta$ of *eDen*, the largest one as shown in Fig. 4(a). The dataset of CoNLL contains a lot of short sentences, such as "Chicago 8,674 484,018" and " SOFIA 1996-12-06". That's why it distinguishes itself from other datasets.

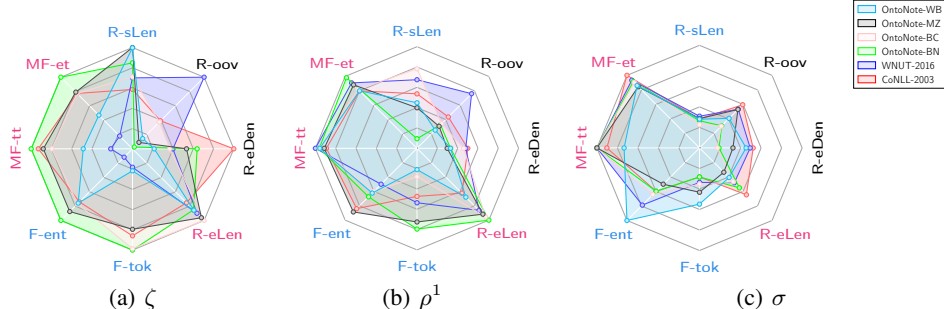

Figure 4: Illustration of dataset biases characterized by task-independent measure $\zeta$, task-dependent measures $\rho^1$ and $\sigma$.

The intrinsic differences in datasets can help us to understand how different datasets influence the different choices of models, which will be explained later (Sec. 5.4).

| Model | F1 | Spearman | | | | | | | | Standard Deviation | | | | | | | | | |
|---|---|---|---|---|---|---|---|---|---|---|---|---|---|---|---|---|---|---|---|
| | | R-eDen | R-oov | R-sLen | MF-et | MF-tt | F-ent | F-tok | R-eLen | R-eDen | R-oov | R-sLen | MF-et | MF-tt | F-ent | F-tok | R-eLen | F-tag | R-mor |
| CRF++ | 80.74 | 60 | **72** | -71 | 80 | 80 | **100** | **96** | -50 | 7.4 | 7.7 | 3.4 | 12 | 9.3 | 9.0 | 4.8 | 7.0 | 6.3 | 19 |
| *CnonWrandlstmCrf* | 78.13 | **67** | 53 | -86 | **100** | **100** | 89 | 89 | -50 | 7.3 | 7.0 | 4.5 | **15** | **17** | **12** | **9.7** | **7.8** | 4.8 | 12 |
| *CcnnWnonelstmCrf* | 77.01 | **67** | 70 | -64 | 60 | 80 | **100** | 82 | -50 | **8.3** | **9.2** | **6.0** | 11 | 7.1 | 6.5 | 2.5 | 6.3 | **6.9** | 16 |
| *CcnnWrandlstmCrf* | 83.80 | **67** | 68 | -89 | **100** | 90 | 89 | 61 | -50 | 6.1 | 5.8 | 2.6 | 10 | 9.8 | 6.9 | 3.4 | 7.3 | 4.7 | **22** |
| *CcnnWglovelstmCrf* | 90.48 | 60 | 40 | -75 | 80 | 90 | 71 | 14 | -50 | 2.9 | 3.5 | 1.6 | 6.7 | 7.1 | 3.3 | 0.6 | 5.8 | 5.2 | 15 |
| *CcnnWglovecnnCrf* | 90.14 | 55 | 48 | -82 | 80 | 90 | 71 | -7.1 | **-100** | 3.4 | 4.2 | 1.7 | 7.1 | 7.1 | 3.2 | 0.9 | 5.9 | 5.3 | 15 |
| *CcnnWglovelstmMlp* | 88.05 | 57 | 42 | -71 | 80 | 90 | 96 | -39 | -50 | 3.2 | 4.9 | 2.0 | 8.2 | 9.4 | 3.9 | 0.9 | 5.3 | 6.7 | 12 |
| *CelmWnonelstmCrf* | 91.64 | 40 | 30 | 50 | 80 | 90 | 96 | 57 | -50 | 2.7 | 2.9 | 1.0 | 5.5 | 4.4 | 2.7 | 0.7 | 4.0 | 5.5 | 10 |
| *CelmWglovelstmCrf* | 92.22 | 45 | 27 | 11 | 80 | 90 | 68 | -32 | **-100** | 2.2 | 3.5 | 0.9 | 5.1 | 4.3 | 2.5 | 1.0 | 3.7 | 5.8 | 15 |
| *CbertWnonelstmMlp* | 91.11 | 38 | 5.0 | 29 | 80 | 90 | 46 | 14 | -50 | 2.9 | 3.5 | 1.6 | 5.9 | 4.5 | 3.2 | 1.3 | 2.5 | 5.6 | 10 |
| *CflairWnonelstmCrf* | 89.98 | 19 | 47 | 0.0 | 60 | 90 | **100** | 46 | -50 | 3.1 | 2.9 | 1.4 | 6.2 | 4.9 | 3.3 | 1.2 | 4.3 | 6.2 | 15 |
| *CflairWglovelstmCrf* | 93.03 | 26 | 22 | -14 | 80 | 90 | 79 | 29 | **-100** | 2.1 | 3.2 | 0.9 | 4.9 | 4.0 | 2.3 | 0.6 | 3.5 | 4.8 | 11 |

Table 4: Model-wise measures $\mathbf{S}_{i,j}^{\rho}$ and $\mathbf{S}_{i,j}^{\sigma}$ on CoNNL-2003. Pink attributes are used to characterize *category ambiguity* of entities while blue attributes can measure the degree to which test entities have been seen in training set.

## 5.3 MODEL-WISE ANALYSIS

Based on two model-wise measures: $\mathbf{S}_{i,j}^{\rho}$ and $\mathbf{S}_{i,j}^{\sigma}$ defined in Eq.3 and Eq.4, we investigate how different attributes influence the performance of the models with different architectures and pre-trained embeddings. Fig. 4 illustrates the case on CoNLL (other datasets are included in appendix), and we have observed that:

1) **Char-unaware models are more sensitive to the degree of category ambiguity and occurrence of entities.** We observe that "*CnonWrandlstmCrf*" is negatively related to MF-et, MF-tt, F-ent and F-tok with high values of $\rho$ and $\sigma$, suggesting the importance of the character-level encoder, which plays a major role in generalization to rare entities and entities with multiple categories. More importantly, this observation still holds on other datasets. (See appendix)

2) **Sentence length is a swing factor, whose contribution depends on what types of pre-trained embeddings are used.** There is a strong negative correlation between models with context-independent embeddings and the attribute R-sLen (sentence length). The relationship is reversed when contextualized embeddings are used. The reason we believe is that long sentences could provide sufficient context information for contextualized models. It is noticeable, however that flair-related models "*CflairWnonelstmCrf*" behave differently compared with other contextualized models, which

we will explain later in Sec 5.4 (Flair performs worse than ELMo when dealing with long sentences due to its structural bias).

3) **Character encoders favor the sentences with high entity-density.** Only using character-level CNN is easy to overfit the feature of capital letters. As a result, more non-entities are mis-predicted as entities. Based on the understanding of the previous analysis (Sec. 5.2) of the entity density's influence on CoNLL, we could better explain why "*CcnnWnonelstmCrf*" achieves the highest value of $\rho$ and $\sigma$ in "*R-eDen* (entity density) attribute in Tab.4.

### 5.4 Bucket-wise Analysis

In this section, we choose several typical models (others are shown in the appendix) as analyzed cases, aiming to seek answers to the following questions: 1) What are the strengths and weakness of different architectural designs? In other words, does a worse-ranked model outperform the best-ranked model in some aspect? 2) How do the different choices of datasets influence model performance? Tab. 5 illustrates the bucket-wise measure $\beta$, which is computed based on any pair of models M1 and M2. Next, we list some of our observations. Others are illustrated in the appendix.

**CNN v.s. LSTM** The sentence encoder of CNN is better at dealing with short sentences, which holds in all datasets we evaluated in this paper. Strikingly, CNN outperforms LSTM by a large margin (dataset-level F1) on `MZ` dataset while is significantly worse than LSTM on `WB` (refer to the appendix.). We attempt to explain these discrepancies based on above attribute-wise metric $\zeta$ in Fig. 4(a): `sentence length` and `entity density` are two major factors for the choices of CNN and LSTM. CNN is better than LSTM when the dataset with higher value of $\zeta_{sLen}$ and $\zeta_{eDen}$. By contrast, when a dataset with lower $\zeta_{eDen}$, LSTM is a priority (The $\zeta_{eDen}$ of `WB` is the lowest).

**CRF v.s. MLP** The benefits of using CRF on short sentences are very stable, and improvement can be seen in all datasets. Similarly, based on attribute-wise metric $\zeta$ in Fig. 4, we find *category ambiguity* (`MF-et`, `MF-tt`) is a major factor for the choices of CRF and MLP: if a dataset with higher $\zeta_{MF-et}$, in which longer entities can benefit more from CRF-based models. In comparison, introducing CRF will lead to more errors on long entities once the dataset (i.e. `BN`, `MZ`) has a lower $\zeta_{MF-et}$.

**CcnnWrand v.s. CcnnWnone** The question has been little studied whether we need an extra word-level method (i.e., word-level look-up table) to get word representations when we have already used CNN to obtain word representations. Here, we show that `CcnnWrand` is not always better than `CcnnWnone` and entity length matters for the choice. Specifically, `CcnnWnone` could achieve better performance on on the `WNUT` and `MZ` datasets. With the help of Fig. 4-(a), we find the two datasets share a property of much higher value of $\zeta_{R-ele}$ (entity length). Additionally, another commonality between the two datasets can be observed from Tab. 5: the gain of `CcnnWnone` mainly comes from the entities with the longer length.

**ELMo v.s GloVe** ELMo consistently outperforms GloVe on all datasets using the holistic F1 score, but it is worse than GloVe when modeling short sentences. Another interesting finding is that for those sentences containing more OOV tokens, GloVe could achieve better performance on all datasets. These phenomena indicate the complementarity between ELMo and GloVe, and our further combination of these two embeddings (*CelmWglovelstmCrf* in Tab. 3) indeed works better.

**Flair v.s. ELMo** While the current state-of-the-art NER model Flair has achieved the best performance in terms of dataset-level F1 score, a worse-ranked model (ELMo) could outperform it in some aspects. Typically, Flair performs worse when dealing with long sentences. The reason can be attributed to the its structure design, which adopts a LSTM-based encoder for character language modeling, suffering from long-term dependency problem (especially for character-level language model). A promising improvement is to use the Transformer-based architecture for character language model.

## 6 Conclusion

To bridge the gap between the insufficient understanding of the nature of datasets and model designs, this paper proposes a generalized evaluation methodology to interpret model biases, dataset biases,

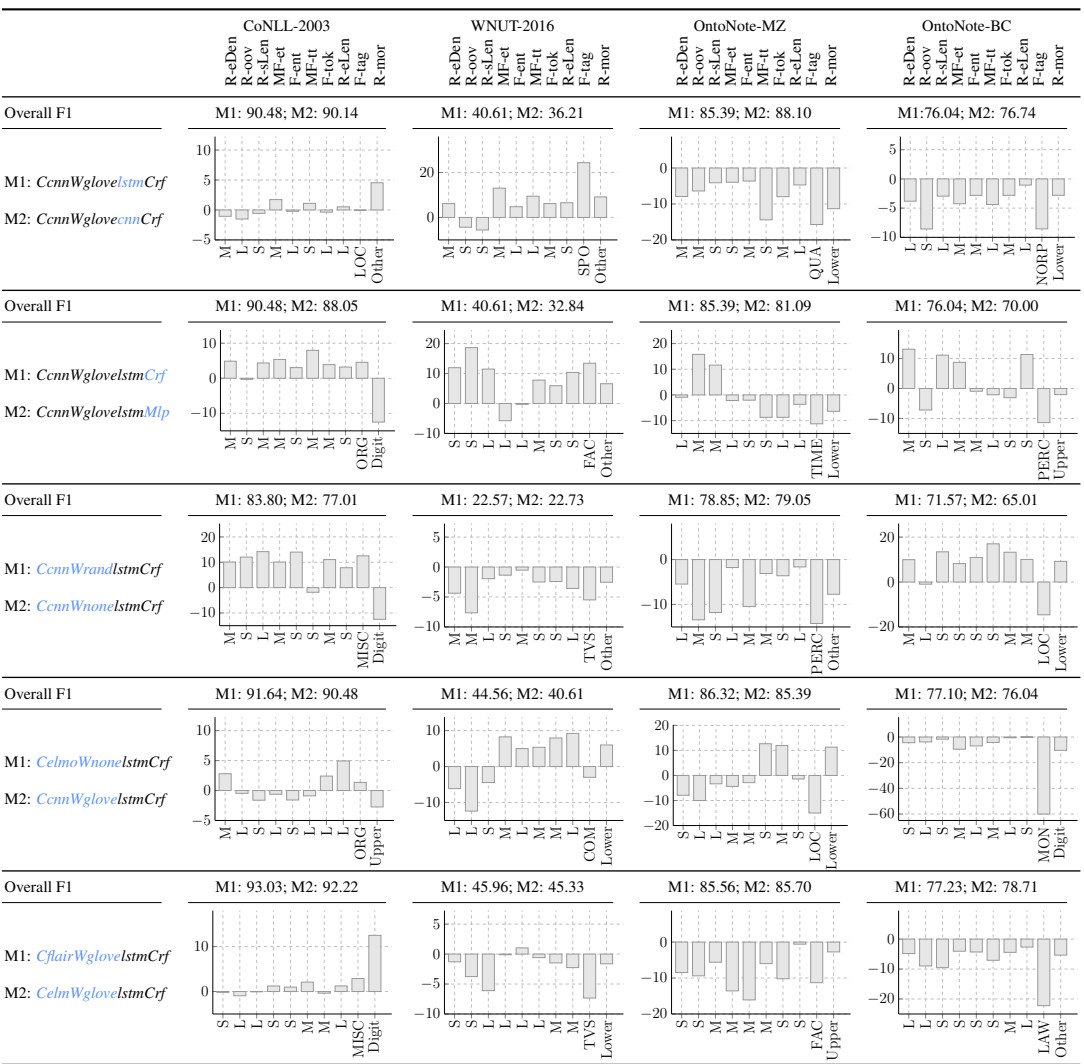

Table 5: Illustration of the bucket-wise measure $\beta$. Each histogram is obtained based on subtracting the performance of Model1 (M1) from Model2 (M2) on a bucket. For ease of presentation, we roughly classify some attribute values into three categories: small(S), middle(M) and large(L). For example, the first column of the top left histogram represents M2 outperforms M1 when the attribute R-eDen takes the small (S) values.

and their correlation, drawing on the complementary strengths of the fine-grained evaluation and multi-dataset evaluation. We choose the NER task as a test case, the observations based on the extensive experiments suggest directions for improvement and can drive the progress of this area.

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

## A   HYPER-PARAMETERS

| Character- / Word-Level | | Sentence-Level and Training | |
| --- | --- | --- | --- |
| **Hyper-parameters** | **Value** | **Hyper-parameters** | **Value** |
| Character emb. size | 25 | LSTM layer | 1 |
| CNN layer | 1 | LSTM hidden size | 100 |
| CNN kernel size | 3 | CNN layer | 2 |
| ELMo model | Original | CNN kernel size | 3 |
| BERT model | bert-base-cased | Learning rate | 0.01 |
| Flair model | forward & backward | Learning rate decay | 0.05 |
| GloVe emb. size | 100 | Batch size | 10 |
| Word random emb. size | 100 | Optimizer | sgd |

Table 6: Hyper-parameters for our evaluated models.

## B   MODEL-WISE ANALYSIS ON THE OTHER DATASETS

| | Spearman | | | | | | | | Standard Deviation | | | | | | | | | |
| --- | --- | --- | --- | --- | --- | --- | --- | --- | --- | --- | --- | --- | --- | --- | --- | --- | --- | --- |
| Model | R-eDen | R-oov | R-sLen | MF-et | F-ent | MF-tt | F-tok | R-eLen | R-eDen | R-oov | R-sLen | MF-et | F-ent | MF-tt | F-tok | R-eLen | F-tag | R-mor |
| CRF++ | -33 | -75 | 82 | 80 | 80 | 100 | 50 | -100 | 6.3 | 7.4 | 5.4 | 16 | 13 | 10 | 2.5 | 4 | 13 | 5.9 |
| *CnonWrandlstmCrf* | -80 | -61 | 57 | 80 | 100 | 100 | 30 | -100 | 4.9 | 6.6 | 4.1 | 25 | 22 | 19 | 6.5 | 7.3 | 10 | 3 |
| *CcnnWnonelstmCrf* | -37 | -69 | 86 | 80 | 100 | 100 | 20 | -100 | 6.2 | 7.5 | 6.3 | 15 | 14 | 9.2 | 2.2 | 5.2 | 14 | 6.8 |
| *CcnnWrandlstmCrf* | -48 | -68 | 61 | 80 | 100 | 100 | 30 | -100 | 5.6 | 7.9 | 3.6 | 18 | 15 | 12 | 3.3 | 7.3 | 14 | 5.6 |
| *CcnnWglovelstmCrf* | -48 | -80 | 86 | 100 | 20 | 100 | -60 | -100 | 9.5 | 12 | 6.2 | 16 | 13 | 22 | 7.3 | 14 | 20 | 12 |
| *CcnnWglovecnnCrf* | -43 | -86 | 21 | 100 | 40 | 100 | -60 | -50 | 9.3 | 11 | 3.2 | 16 | 12 | 20 | 7 | 12 | 22 | 10 |
| *CcnnWglovelstmMlp* | -62 | -81 | 75 | 100 | 20 | 100 | -60 | -50 | 9.4 | 9.8 | 4.2 | 20 | 16 | 24 | 6.8 | 10 | 19 | 9.9 |
| *CelmWnonelstmCrf* | -52 | -70 | 68 | 100 | 20 | 100 | -70 | -100 | 13 | 12 | 8.3 | 15 | 13 | 22 | 7.8 | 10 | 19 | 12 |
| *CelmWglovelstmCrf* | -47 | -79 | 43 | 100 | 20 | 100 | -70 | -100 | 12 | 11 | 7 | 16 | 13 | 22 | 7.2 | 9.1 | 20 | 11 |
| *CbertWnonelstmMlp* | -47 | -84 | 79 | 80 | 20 | 100 | -70 | -50 | 12 | 12 | 5.7 | 17 | 14 | 21 | 9.4 | 9.4 | 22 | 13 |
| *CflairWnonelstmCrf* | -50 | -74 | 39 | 80 | 40 | 100 | -70 | -100 | 12 | 9.7 | 7 | 16 | 13 | 21 | 7.4 | 8.9 | 23 | 12 |
| *CflairWglovelstmCrf* | -32 | -81 | 86 | 100 | 32 | 100 | -70 | -100 | 11 | 11 | 8.5 | 16 | 13 | 19 | 8 | 8.8 | 21 | 10 |

Table 7: Model-wise measures $\mathbf{S}^{\rho}_{i,j}$ and $\mathbf{S}^{\sigma}_{i,j}$ on Wnut-16.

| | Spearman | | | | | | | | Standard Deviation | | | | | | | | | |
| --- | --- | --- | --- | --- | --- | --- | --- | --- | --- | --- | --- | --- | --- | --- | --- | --- | --- | --- |
| Model | R-eDen | R-oov | R-sLen | MF-et | F-ent | MF-tt | F-tok | R-eLen | R-eDen | R-oov | R-sLen | MF-et | F-ent | MF-tt | F-tok | R-eLen | F-tag | R-mor |
| CRF++ | 43 | 40 | -18 | 100 | 90 | 98 | 86 | -100 | 2.3 | 3.8 | 3.9 | 13 | 8.6 | 13 | 4.8 | 6.8 | 25 | 7.3 |
| *CnonWrandlstmCrf* | -70 | 20 | -21 | 100 | 86 | 98 | 86 | -100 | 2.1 | 5.7 | 1.8 | 18 | 12 | 22 | 7.5 | 7.0 | 18 | 9.0 |
| *CcnnWnonelstmCrf* | 5.0 | -40 | 0.0 | 80 | 98 | 88 | 75 | -100 | 2.3 | 3.5 | 3.5 | 14 | 10 | 14 | 4.9 | 8.1 | 25 | 7.3 |
| *CcnnWrandlstmCrf* | 15 | 40 | 0.0 | 100 | 71 | 100 | 86 | -100 | 2.5 | 4.3 | 2.8 | 14 | 8.9 | 14 | 4.1 | 8.1 | 18 | 5.8 |
| *CcnnWglovelstmCrf* | -12 | -40 | 11 | 100 | 76 | 98 | 86 | -100 | 2.3 | 2.5 | 2.7 | 9.4 | 6.0 | 11 | 1.9 | 7.6 | 17 | 6.3 |
| *CcnnWglovecnnCrf* | -27 | -40 | 0.0 | 100 | 62 | 95 | 54 | -100 | 2.6 | 2.8 | 4.6 | 9.4 | 6.1 | 11 | 2.0 | 8.2 | 18 | 4.9 |
| *CcnnWglovelstmMlp* | -82 | 40 | 0.0 | 100 | 90 | 93 | 75 | -100 | 2.7 | 3.5 | 3.5 | 12 | 7.8 | 13 | 3.5 | 5.8 | 22 | 7.6 |
| *CelmWnonelstmCrf* | 43 | 0.0 | -11 | 100 | 24 | 98 | 71 | -100 | 1.8 | 2.9 | 2.3 | 6.8 | 4.7 | 8.0 | 2.1 | 5.5 | 16 | 6.6 |
| *CelmWglovelstmCrf* | 30 | -40 | -14 | 100 | 40 | 98 | 79 | -100 | 2.0 | 4.3 | 4.2 | 7.4 | 4.6 | 8.6 | 2.0 | 5.4 | 12 | 6.1 |
| *CbertWnonelstmMlp* | 17 | -20 | 7.1 | 100 | 29 | 98 | 86 | -100 | 2.1 | 3.8 | 2.6 | 7.7 | 4.6 | 8.0 | 2.0 | 3.4 | 15 | 4.7 |
| *CflairWnonelstmCrf* | -18 | 40 | -21 | 100 | 76 | 95 | 93 | -100 | 2.4 | 3.2 | 3.0 | 8.8 | 5.6 | 10 | 2.6 | 6.8 | 18 | 7.1 |
| *CflairWglovelstmCrf* | -13 | -40 | 21 | 100 | 55 | 95 | 86 | -100 | 1.5 | 3.5 | 2.3 | 8.9 | 5.2 | 9.2 | 2.3 | 6.3 | 17 | 6.0 |

Table 8: Model-wise measures $\mathbf{S}^{\rho}_{i,j}$ and $\mathbf{S}^{\sigma}_{i,j}$ on OntoNote-BN.

## C   BUCKET-WISE ANALYSIS

| Model | Spearman | | | | | | | | Standard Deviation | | | | | | | | | |
|---|---|---|---|---|---|---|---|---|---|---|---|---|---|---|---|---|---|---|
| | R-eDen | R-oov | R-sLen | MF-et | F-ent | MF-tt | F-tok | R-eLen | R-eDen | R-oov | R-sLen | MF-et | F-ent | MF-tt | F-tok | R-eLen | F-tag | R-mor |
| CRF++ | -30 | 14 | -46 | 100 | 71 | 95 | 29 | -100 | 2.6 | 4.9 | 3.9 | 19 | 13 | 18 | 5.7 | 8.3 | 26 | 14 |
| *CnonWrandlstmCrf* | -55 | 1.3 | -57 | 80 | 43 | 93 | 67 | -100 | 3.5 | 5 | 2.5 | 21 | 14 | 23 | 8.6 | 7.2 | 25 | 17 |
| *CcnnWnonelstmCrf* | -52 | 37 | -71 | 80 | 89 | 98 | 57 | -100 | 3.6 | 5.5 | 3.7 | 18 | 13 | 19 | 5.9 | 8.3 | 26 | 12 |
| *CcnnWrandlstmCrf* | -63 | 24 | -86 | 80 | 75 | 98 | 52 | -100 | 3.6 | 5.4 | 4.6 | 17 | 12 | 18 | 5.8 | 10 | 25 | 11 |
| *CcnnWglovelstmCrf* | -48 | 52 | -89 | 100 | 61 | 98 | 0 | -100 | 3.7 | 5.6 | 5 | 13 | 9.1 | 16 | 5.1 | 8.8 | 24 | 11 |
| *CcnnWglovecnnCrf* | -8.3 | 39 | -96 | 80 | 64 | 93 | -12 | -100 | 2.9 | 4.5 | 4.9 | 12 | 9.4 | 16 | 5.6 | 8.6 | 25 | 11 |
| *CcnnWglovelstmMlp* | -35 | 43 | -96 | 100 | 50 | 100 | -7.1 | -50 | 4.3 | 5.7 | 5.8 | 15 | 12 | 19 | 8.4 | 7.7 | 24 | 13 |
| *CelmWnonelstmCrf* | 67 | 64 | -96 | 60 | 46 | 98 | 12 | -100 | 1.7 | 5.6 | 4.5 | 15 | 9.2 | 14 | 4.3 | 6.6 | 18 | 11 |
| *CelmWglovelstmCrf* | 18 | 48 | -89 | 60 | 39 | 98 | -38 | -100 | 1.7 | 4.7 | 3.4 | 13 | 8.3 | 13 | 5.3 | 6.3 | 17 | 9.3 |
| *CbertWnonelstmMlp* | 23 | 32 | -82 | 80 | 43 | 98 | -7.1 | -50 | 2.2 | 3.7 | 3.1 | 13 | 7.9 | 13 | 5 | 4.3 | 19 | 9 |
| *CflairWnonelstmCrf* | -85 | 33 | -89 | 80 | 54 | 100 | 12 | -100 | 2.2 | 4.3 | 1.2 | 11 | 8.7 | 13 | 5.2 | 6.9 | 22 | 6 |
| *CflairWglovelstmCrf* | -43 | 22 | -43 | 80 | 64 | 88 | 19 | -100 | 2.2 | 4.2 | 2.7 | 12 | 9.7 | 15 | 5.5 | 7 | 22 | 9.1 |

Table 9: Model-wise measures $\mathbf{S}_{i,j}^{\rho}$ and $\mathbf{S}_{i,j}^{\sigma}$ on OntoNote-BC.

| Model | Spearman | | | | | | | | Standard Deviation | | | | | | | | | |
|---|---|---|---|---|---|---|---|---|---|---|---|---|---|---|---|---|---|---|
| | R-eDen | R-oov | R-sLen | MF-et | F-ent | MF-tt | F-tok | R-eLen | R-eDen | R-oov | R-sLen | MF-et | F-ent | MF-tt | F-tok | R-eLen | F-tag | R-mor |
| CRF++ | -44 | -42 | 14 | 80 | 100 | 93 | 64 | -100 | 6.2 | 10 | 5.5 | 15 | 10 | 17 | 10 | 3.5 | 31 | 7.9 |
| *CnonWrandlstmCrf* | -27 | -25 | 11 | 100 | 89 | 88 | 93 | -100 | 4.7 | 14 | 4.1 | 21 | 12 | 25 | 11 | 5.5 | 28 | 9.1 |
| *CcnnWnonelstmCrf* | -10 | -32 | 21 | 80 | 77 | 93 | 64 | -50 | 5.1 | 8.1 | 5.6 | 13 | 7.5 | 16 | 6.8 | 6.3 | 30 | 5 |
| *CcnnWrandlstmCrf* | -53 | -43 | 71 | 80 | 94 | 90 | 74 | -100 | 4.3 | 8.3 | 5.5 | 13 | 9.2 | 15 | 8 | 6.8 | 30 | 4.9 |
| *CcnnWglovelstmCrf* | -60 | 0 | 64 | 80 | 89 | 95 | 71 | -100 | 4.6 | 5.1 | 2.7 | 8.8 | 5.8 | 15 | 4.3 | 7.6 | 28 | 8.6 |
| *CcnnWglovecnnCrf* | -38 | -23 | 68 | 100 | 77 | 90 | 71 | -100 | 4.2 | 5.4 | 3 | 7.9 | 5.2 | 12 | 4.1 | 6.4 | 27 | 5 |
| *CcnnWglovelstmMlp* | -43 | -45 | 64 | 100 | 60 | 95 | 64 | -100 | 4.9 | 7.7 | 2.9 | 13 | 7.3 | 13 | 5.2 | 2.5 | 29 | 5.4 |
| *CelmWnonelstmCrf* | -17 | -37 | 57 | 100 | 94 | 95 | 71 | -100 | 4.6 | 6.5 | 2.4 | 9.6 | 5.6 | 12 | 4.6 | 4.6 | 28 | 5.7 |
| *CelmWglovelstmCrf* | -37 | -50 | 11 | 100 | 100 | 95 | 75 | -100 | 5.1 | 7.3 | 5.4 | 10 | 5.7 | 13 | 4.1 | 4.9 | 26 | 6.6 |
| *CbertWnonelstmMlp* | -6.7 | -4.5 | 21 | 80 | 83 | 83 | 79 | -50 | 4.5 | 5.6 | 3.6 | 11 | 6 | 10 | 4.9 | 2.1 | 26 | 4.6 |
| *CflairWnonelstmCrf* | 15 | -30 | 14 | 80 | 100 | 90 | 71 | -100 | 4 | 7 | 5.1 | 13 | 6.5 | 13 | 5.6 | 3.6 | 26 | 5.5 |
| *CflairWglovelstmCrf* | -6.7 | -37 | 61 | 80 | 100 | 86 | 71 | -100 | 4.4 | 6.8 | 3.9 | 12 | 6.1 | 12 | 5.6 | 4.4 | 26 | 5.2 |

Table 10: Model-wise measures $\mathbf{S}_{i,j}^{\rho}$ and $\mathbf{S}_{i,j}^{\sigma}$ on OntoNote-MZ.

| Model | Spearman | | | | | | | | Standard Deviation | | | | | | | | | |
|---|---|---|---|---|---|---|---|---|---|---|---|---|---|---|---|---|---|---|
| | R-eDen | R-oov | R-sLen | MF-et | F-ent | MF-tt | F-tok | R-eLen | R-eDen | R-oov | R-sLen | MF-et | F-ent | MF-tt | F-tok | R-eLen | F-tag | R-mor |
| CRF++ | -20 | 43 | -21 | 80 | 50 | 90 | 8.6 | -100 | 8.1 | 11 | 9.3 | 22 | 22 | 22 | 12 | 5 | 20 | 8.2 |
| *CnonWrandlstmCrf* | -55 | 0 | 54 | 80 | 54 | 98 | -26 | -100 | 13 | 11 | 9.5 | 27 | 21 | 27 | 16 | 12 | 21 | 8.7 |
| *CcnnWnonelstmCrf* | -37 | -4.8 | -11 | 80 | 64 | 93 | -2.9 | -50 | 11 | 6.1 | 8.5 | 20 | 20 | 21 | 13 | 9.2 | 21 | 8.5 |
| *CcnnWrandlstmCrf* | -33 | 7.1 | 3.6 | 80 | 61 | 93 | -37 | -100 | 11 | 12 | 4.6 | 24 | 20 | 27 | 15 | 13 | 23 | 8.7 |
| *CcnnWglovelstmCrf* | -32 | 40 | -86 | 80 | 50 | 98 | -37 | -100 | 11 | 11 | 5.6 | 18 | 14 | 22 | 12 | 9.9 | 23 | 9.9 |
| *CcnnWglovecnnCrf* | -32 | -2.4 | -63 | 80 | 54 | 81 | -43 | -50 | 11 | 13 | 5.1 | 22 | 18 | 26 | 17 | 13 | 24 | 13 |
| *CcnnWglovelstmMlp* | -40 | 33 | -71 | 80 | 68 | 95 | -14 | -50 | 11 | 9.2 | 5.2 | 21 | 17 | 26 | 16 | 11 | 23 | 13 |
| *CelmWnonelstmCrf* | -37 | 29 | -39 | 80 | 64 | 98 | -43 | -50 | 11 | 9.9 | 5.8 | 17 | 14 | 23 | 10 | 8.4 | 24 | 9.2 |
| *CelmWglovelstmCrf* | -27 | 45 | -14 | 80 | 71 | 98 | -26 | -50 | 9.7 | 10 | 3.1 | 17 | 13 | 21 | 11 | 8.8 | 26 | 9.5 |
| *CbertWnonelstmMlp* | -25 | 2.4 | -54 | 80 | 71 | 98 | -2.9 | -50 | 8.7 | 5.1 | 5.7 | 16 | 13 | 20 | 12 | 7.1 | 28 | 10 |
| *CflairWnonelstmCrf* | -30 | 24 | -79 | 80 | 71 | 95 | -26 | -50 | 10 | 11 | 7 | 18 | 16 | 21 | 10 | 8.9 | 24 | 8.9 |
| *CflairWglovelstmCrf* | -28 | 50 | -61 | 80 | 61 | 98 | 8.6 | -50 | 10 | 6.2 | 5.8 | 16 | 14 | 21 | 10 | 9.2 | 24 | 11 |

Table 11: Model-wise measures $\mathbf{S}_{i,j}^{\rho}$ and $\mathbf{S}_{i,j}^{\sigma}$ on OntoNote-WB.

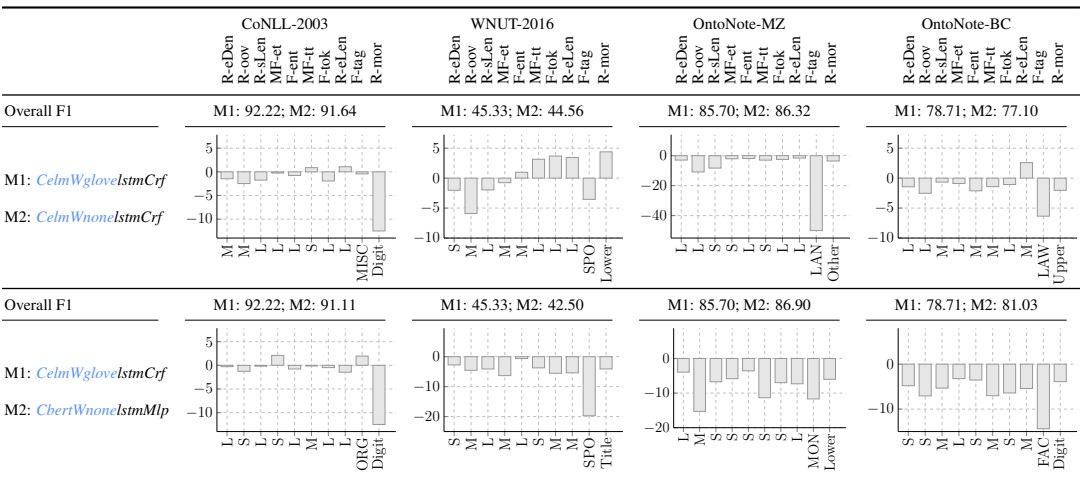

Table 12: A supplement bucket-wise analysis results to Tab. 5. Each histogram is obtained based on subtracting the performance of Model1 (M1) from Model2 (M2) on a bucket. For ease of presentation, we roughly classify some attribute values into three categories: small(S), middle(M) and large(L). For example, the first column of the top left histogram represents M2 outperforms M1 when the attribute R-eDen takes the small (S) values.

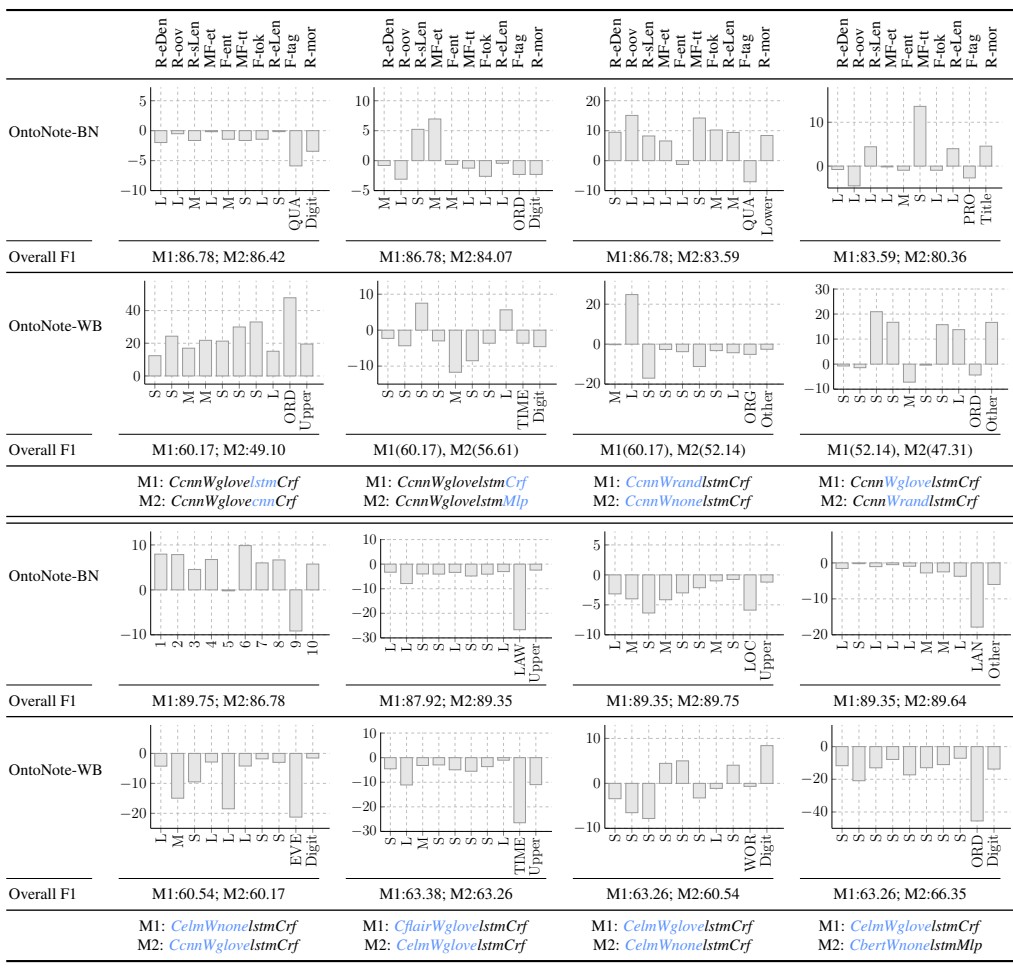

Table 13: Illustration of the bucket-wise measure $\beta$ on OntoNote-BN and OntoNote-WB. Each histogram is obtained based on subtracting the performance of Model1 (M1) from Model2 (M2) on a bucket. For ease of presentation, we roughly classify some attribute values into three categories: small(S), middle(M) and large(L).

