# OpenReview forum: "Towards Interpretable Evaluations: A Case Study of Named Entity Recognition"
_ICLR.cc/2020/Conference — Reject_

### Official Review · AnonReviewer3 · 2019-10-22
**Official Blind Review #3**

**Rating:** 3

**Review:**


The manuscript proposes an evaluation methodology to obtain deeper insights regarding the strength and weaknesses of different methods on different datasets. The method considers a set of methods addressing the task of Named Entity Recognition (NER) as case study. In addition, it proposes a set of attribute-based criteria, i.e. bucketization strategies, under which the dataset can be divided and analyzed in order to highlight different properties of the evaluated methods.

As said earlier, the manuscript proposes an evaluation methodology to obtain deeper insights regarding the strength and weaknesses of different methods on different datasets. The characteristic of being able to provided deeper insights on strength/weaknesses and relevant factors on the inner-workings of a given method is
something very desirable for every evaluation. As such, in my opinion, the "interpretable" tag associate to the proposed method is somewhat out of place. Having said that, I would recommend removing the "interpretable" tag and stress the contribution of this manuscript as an evaluation protocol.

In Section 4.2, for the R-Bucket strategy it is stated as having the requirement of discrete and finite attributes. Based on the equations of the other two strategies (R-bucket and F-bucket), it seems that they also have the requirement of having discrete attributes. Is this indeed the case? if so, it should be explicitly indicated.
Having said that, this raises another question: Is this protocol exclusive to tasks/problems with explicit discrete attributes?

The goal of this manuscript is to propose a general evaluation protocol for NLP tasks.
However, it seems to be somewhat tailored to the NER task. My question is: How well the proposed method generalizes to other NLP tasks without attributes? Similarly, how well the proposed bucketization strategies generalize beyond the NER task? Perhaps the generalization characteristics and limitations of the proposed evaluation methodology should be explicitly discussed in the manuscript.

Last paragraph of Section 4.2 summarizes ideas that were just presented. It feels somewhat redundant. I suggest removing in in favor of extending the existing discussions and analysis.

I may consider upgrading my initial rating based on on the feedback given to my questions/doubts.


**Experience Assessment:**

I have published one or two papers in this area.

**Review Assessment: Checking Correctness Of Derivations And Theory:**

I assessed the sensibility of the derivations and theory.

**Review Assessment: Checking Correctness Of Experiments:**

I assessed the sensibility of the experiments.

**Review Assessment: Thoroughness In Paper Reading:**

I read the paper thoroughly.

---

> ### Author Response · Authors · 2019-11-14
> **Response to Review #3-Part1**
>
>
> *****************************************************************************************
> Tasks			                        Attributes			                     Measures		Related Ref.
> *****************************************************************************************
> Machine Translation	               sentence length			             Bleu		              [1]
> Machine Translation	               word (or N-gram) frequency	     Accuracy*		      [2]
> 			                               in the training set.
> Machine Translation	               word POS-tag in the training set    Accuracy*		       [3]
> Machine Translation	               words in reference file		             Word likelihood      [3]
> ------------------------------------------------------------------------------------------------------------------------
> Summarization (Ext. or Abs.)	sentence length			             Rouge		        -
> Summarization (Ext. or Abs.)	compression of summary		     Rouge		       [6]
> Summarization (Ext. or Abs.)	density of summary		             Rouge		       [6]
> Summarization (Ext. or Abs.)	volume overlap			             Rouge		       [5]
> ------------------------------------------------------------------------------------------------------------------------
> Summarization (Ext.)	                 position of each sentence		     Rouge/Accuracy     [4] [5]
> Summarization (Ext.)	                 OOV rate of sentence		     Rouge		         -
> ------------------------------------------------------------------------------------------------------------------------
> Semantic Matching		         length of sent1 or sent2	             Accuracy		        [7]
> Semantic Matching		         Func(sent1, sent2)	                     Accuracy	                  -
> Semantic Matching		         OOV				                     Accuracy		          -
> ------------------------------------------------------------------------------------------------------------------------
> QA			                                 answer length, type, position	    Matching F1	         [8]
> QA			                                 document length			    Matching F1	         [12]
> QA			                                 query length, type			    Matching F1	         [9]
> ------------------------------------------------------------------------------------------------------------------------
> Text Classification		         sentence/word length		     Accuracy		          [11]
> Text Classification		         OOV				                     Accuracy		          [10]
> Text Classification		         sentence familiarity			     Accuracy		           -
> ------------------------------------------------------------------------------------------------------------------------
> Sequence labeling		         Similar to this work
> ------------------------------------------------------------------------------------------------------------------------
>
> 【Footnotes】
> "Accuracy*" :  whether generated words appeared in the gold reference
> “Func” can be used to compute sentence length difference.
> “Sentence familiarity”: we could quantify the degree to which the test sentence has been seen in the training set (based on n-gram calculation).
>
> 【References】
> [1] Effective Approaches to Attention-based Neural Machine Translation, Minh-Thang Luong Hieu Pham
> Christopher D. Manning
> [2] Von misesfisher loss for training sequence to sequence, Sachin Kumar and Yulia Tsvetkov.
> [3] Compare-mt: A Tool for Holistic Comparison of Language Generation Systems, Graham Neubig,
> Zi-Yi Dou, Junjie Hu, Paul Michel, Danish Pruthi, Xinyi Wang, John Wieting
> [4] Text Summarization with Pretrained Encoders, Yang Liu,  Mirella Lapata
> [5] Earlier Isn’t Always Better: Sub-aspect Analysis on Corpus and System Biases in Summarization,
> Taehee Jung, Dongyeop Kang, Lucas Mentch, Eduard Hovy
> [6] A Closer Look at Data Bias in Neural Extractive Summarization Models, Ming Zhong, Danqing Wang,
> Pengfei Liu, Xipeng Qiu, Xuanjing Huang
> [7] Improved Semantic Representations From Tree-Structured Long Short-Term Memory Networks, Kai Sheng
> Tai, Richard Socher, Christopher D. Manning
> [8] Bidirectional Attention Flow for Machine Comprehension, Minjoon Seo, Aniruddha Kembhavi, Ali Farhadi, Hannaneh Hajishirzi
> [9] A Thorough Examination of the CNN/Daily Mail Reading Comprehension Task, Danqi Chen, Jason Bolton, Christopher D. Manning
> [10] Learning Semantic Representations of Users and Products for Document Level Sentiment Classification, Duyu Tang, Bing Qin, Ting Liu
> [11] The Relationship of Word Length and Sentence Length: The Inter-Textual Perspective, Peter Grzybek, Ernst
> Stadlober, Emmerich Kelih
> [12] TriviaQA: A Large Scale Distantly Supervised Challenge Dataset for Reading Comprehension, Mandar Joshi, Eunsol Choi, Daniel S. Weld, Luke Zettlemoyer

---

> ### Author Response · Authors · 2019-11-14
> **Response to Review #3-Part2**
>
> Q1: “However, it seems to be somewhat tailored to the NER task. My question is: How well the proposed method generalizes to other NLP tasks without attributes? Similarly, how well the
> proposed bucketization strategies generalize beyond the NER task?”
> A1: To adapt this methodology to other tasks, we honestly admit that the process of attribute definition usually requires some domain knowledge. However, we would like to show the process is not complicated since many task-agnostic attributes could be applied, such as oov, sentence length. Importantly, we believe each domain-specific expert should take responsibility for driving the development of the domain based on their understanding of the task, and hopefully, this work could provide such a methodology in which domain knowledge from different tasks could be utilized.
> As a preliminary summary in the  table (Part1), we share some task-specific definition of attributes, where “Tasks” represents different types of tasks; “Attributes” denotes the criterion that we use to divide the test set, and “Measures” represents the measure we use to evaluate each divided sub-set. “Related ref.” shows the corresponding papers that have adopted the attribute for fine-grained evaluation.
>
>
> Q2: In Section 4.2, for the R-Bucket strategy it is stated as having the requirement of discrete and finite attributes. Based on the equations of the other two strategies (R-bucket and F-bucket), it seems that they also have the requirement of having discrete attributes. Is this indeed the case? if so, it should be explicitly indicated. Having said that, this raises another question: Is this protocol exclusive to tasks/problems with explicit discrete attributes?
> A2: We are sorry for not making it clear, and we have refined the description of the bucket strategy more clearly. Specifically, R- and F- Bucket strategies could be applied to
> attributes with discrete value and continuous value. For example, the “entity density” attribute we used in this paper. Although its value is continuous, we could discretize the value into different ranges (i.e. low, medium, high), and then we could adopt the R- or F-Bucket strategies.
>
>
> Q3: “Last paragraph of Section 4.2 summarizes ideas that were just presented. It feels somewhat
> redundant. I suggest removing in in favor of extending the existing discussions
> and analysis.”
> A3: Thanks for your granular suggestions, and you can see the modification in our revised version.
>
> Q4: "something very desirable for every evaluation. As such, in my opinion, the "interpretable" tag associate to the proposed method is somewhat out of place. Having said that, I would recommend removing the "interpretable" tag and stress the contribution of this manuscript as
> an evaluation protocol."
> A4: Thanks for your constructive suggestion, and we have carefully considered it. However, we have not taken it yet in our revised version, and we would like to share our reasons:
>   1) In this work, we aim to interpret the model biases, dataset biases, and their correlation. Some of the previous work also involves the "bucketize-then-evaluate" idea (As we have listed in the above table and mentioned in the introduction section) while they are without a quantitative process to analyze these biases.
>   2) We also would like to show that this attribute-aided evaluation method could be a way for us to understand our black-box models and datasets.
>
> Thanks again for your insightful comments! We have already refined the paper, please check the latest version. Hope we answer your questions correctly and look forward to your feedback again!

---

### Official Review · AnonReviewer1 · 2019-10-23
**Official Blind Review #1**

**Rating:** 3

**Review:**

TOWARDS INTERPRETABLE EVALUATIONS A CASE STUDY OF NAMED ENTITY RECOGNITION



The authors propose an evaluation methodology to study the relations between datasets and machine learning models. This methodology introduces the notion of attributes which describes different aspects of the samples and buckets which group samples according to the attributes. The goal is to give a better understanding of the strengths and weaknesses of an algorithm on a specific dataset according to the attributes, as shown on Fig4.

The article is very dense and the author chose to present the method from an abstract and generic point of view which makes the reading of the article difficult. In the end, the proposition is a formalisation of the simple error analysis which is commonly done when trying to improve a machine learning system. The advantage of the method could be to introduce some metrics to make the error analysis more automatic. These metrics are given in section 4 but here again only from a formal point of view : it is very difficult for the reader to understand how to interpret them and how to use them for a practical case.

The paper is 17 pages long with the annex : it would better fit a journal publication or the author should select some of the main results to present them in a conference paper. The aspects of the paper related to learning relations is no put forward enough.

2. Related work

2.1 :
 -supplementary exam : unclear

2.2 :
- methodological perspective : a bit a repetition of introduction
- task perspective : not very clear, is the main message  "it important to understand what in the dataset make the model work ?"

3 Task

Section is too small to be a level 1 title

4 Attributes

figure 2 : where are the links to levels ?

4.2 :

  * familiarity : test/train distribution should be the same. Fk computer on train set because it is bigger ? it allows to study the impact of the number of occurrences in the training set. Is it more interesting than a learning curve ?

  * multi attribute familiarity : risk of metric explosion ? how to select the attributes ?

  * eq 3 : spearman not defined

* 4.3

  * metric are defined by formula but it is difficult to understand what is the rationale behind each of them and therefore figure out how to interpret them

  * "Usually where a, b represent two different models and usually model a has a higher performance (by dataset-level metric)" : unclear

5 Experimental setting

Table3 :
the encoding of the model name is not clear
a metric on all the dataset for each model could be computed to decide which one is the best overall
how did you choose the tested combinaisons ?

6\.2

analysis of Fig 4 : R-eLen does not existe (R-Ele). what is eta ?

table 4 : spearman\**r*\* ?

6\.4

* CRF vs MLP : "... a major factor for the choices of CRF and MLP: **if** a dataset with higher ζMF−et, in which longer entities can benefit more from CRF-based models." > missing words ?

Writing :

* "Concretely" isn't very natural at the beginning of sentences, same thing with "Formally", 'Intuitively' …

* in 4.1 : "We refer to E, P, K as the sets of entities (i.e. New York), entity attributes (i.e. entity length) and attributes values (i.e. 2)." => "We refer to the sets of entities (i.e. New York) as E, entity attributes (i.e. entity length) as P and attributes values (i.e. 2) as K" would be better

* same thing in 4.3 "we refer to M = m1,··· ,m|M| as a set of **models** and P = p1,··· ,p|P| as a set of **attributes**" doesn't really work, "M = m1,··· ,m|M| is a set of **models** and P = p1,··· ,p|P| is a set of **attributes**" maybe

* in 4.2 page 5 : "the familiarity Fk (p1 , p2 ) is a measure with intriguing explanation …" : not clear

* 6.3 (3) "Only using character-level CNN is apt to overfit the feature of capital letters." **apt** doesnt work here

**Experience Assessment:**

I have read many papers in this area.

**Review Assessment: Checking Correctness Of Derivations And Theory:**

I assessed the sensibility of the derivations and theory.

**Review Assessment: Checking Correctness Of Experiments:**

I assessed the sensibility of the experiments.

**Review Assessment: Thoroughness In Paper Reading:**

I read the paper at least twice and used my best judgement in assessing the paper.

---

> ### Author Response · Authors · 2019-11-14
> **Response to Review #1-Part1**
>
> *****************************************************************************************
> Tasks			                        Attributes			                     Measures		Bucket Strategy
> *****************************************************************************************
> Machine Translation	               sentence length			             Bleu		              R-Buck
> Machine Translation	               word (or N-gram) frequency	     Accuracy*		      R-Buck
> 			                               in the training set.
> Machine Translation	               word POS-tag in the training set    Accuracy*		       R-Buck
> Machine Translation	               words in reference file		             Word likelihood      R-Buck
> ------------------------------------------------------------------------------------------------------------------------
> Summarization (Ext. or Abs.)	sentence length			             Rouge		        R-Buck
> Summarization (Ext. or Abs.)	compression of summary		     Rouge		        R or F-Buck
> Summarization (Ext. or Abs.)	density of summary		             Rouge		        R or F-Buck
> Summarization (Ext. or Abs.)	volume overlap			             Rouge		        R-Buck
> ------------------------------------------------------------------------------------------------------------------------
> Summarization (Ext.)	                 position of each sentence		     Rouge/Accuracy     F-Buck
> Summarization (Ext.)	                 OOV rate of sentence		     Rouge		        R-Buck
> ------------------------------------------------------------------------------------------------------------------------
> Semantic Matching		         length of sent1 or sent2	             Accuracy		        R-Buck
> Semantic Matching		         Func(sent1, sent2)	                     Accuracy	                R-Buck
> Semantic Matching		         OOV				                     Accuracy		        R-Buck
> ------------------------------------------------------------------------------------------------------------------------
> QA			                                 answer length, type, position	    Matching F1	         F-Buck
> QA			                                 document length			    Matching F1	         R-Buck
> QA			                                 query type			    Matching F1	         F-Buck
> ------------------------------------------------------------------------------------------------------------------------
> Text Classification		         sentence/word length		     Accuracy		          R-Buck
> Text Classification		         OOV				                     Accuracy		          R-Buck
> Text Classification		         sentence familiarity			     Accuracy		          F-Buck
> ------------------------------------------------------------------------------------------------------------------------
> Sequence labeling		         Similar to this work
> ------------------------------------------------------------------------------------------------------------------------
>
> Similar to the Table mentioned in R3, the “Tasks” column shows different types of tasks.
> “Attributes” denotes the criterion that we use to divide the test set, and “Measures” represents the measure we use to evaluate each divided sub-set. “Bucket Strategy” shows which types of bucketization methods could be adopted.

---

> ### Author Response · Authors · 2019-11-14
> **Response to Review #1-Part2**
>
> We appreciate your thorough review and helpful suggestions. We will try to address your questions below.
>
> Q1: “These metrics are given in section 4 but here again only from a formal point of view: it is very difficult for the reader to understand how to interpret them and how to use them for a practical case.”
> A1: Thanks for your feedback. Here, it is the generality of the methodology that allows us to
> describe it from a formal point of view. In a practical case, this method could be adapted to other tasks easily. Below, we would like to give a more specific explanation of how to use them for a practical case.
> As shown in the Table (Part1), for a given NLP task, once we determine related attributes and Bucket Strategy, we could calculate proposed metrics in Sec.4.2 (Sec.3.2 in new version) and make similar analyses.
>
> Q2: “In the end, the proposition is a formalisation of the simple error analysis which is commonly
> done when trying to improve a machine learning system. The advantage of the
> method could be to introduce some metrics to make the error analysis more
> automatic.”
> A2: Our method shares some common properties with error analysis, but beyond it:
>     1) Regarding model analysis, (automatic) error analysis suffers from the confirmation
> bias (Tab.1) problem while our method doesn’t.
> Moreover, the development of error analysis stopped at focusing solely on a
> single dataset [1][2][3]. Many challenges will come when we take the multi-dataset setting into account. This work takes a step towards diagnosing the strengths and weaknesses of different models under different datasets.
>     2) Regarding dataset analysis, the proposed methodology enables us to quantify the data biases, knowing more about the characteristics of each dataset, which is beyond the grasp of error analysis.
>
> [1] Joke Daems, Lieve Macken, and Sonia Vandepitte. On the origin of errors: A fine-grained analysis of mt and pe errors and their relationship
> [2] Jonathan K. Kummerfeld and Dan Klein. Error-driven analysis of challenges in coreference resolution
> [3] Jonathan K. Kummerfeld, David Hall, James R. Curran, and Dan Klein. Parser showdown at the wall street corral: An empirical investigation of error types in parser output
>
>
> Q3: “familiarity : test/train distribution should be the same.“
> A3: We’re sorry for not understanding your statement clearly “test/train distribution should be the
> same.” Do you mean if the calculation of the familiarity requires that test/train
> distribution should be the same?  If, in that case, the answer is the calculation of the familiarity doesn’t require that.
>
> Q4: “Fk computer on train set because it is bigger ?”
> A4: Do you mean if we calculate F_k on the train set because it is bigger?
> If, in that case, the answer is no. The F_k is defined over training set is why we called it as #familiarity#: it could reflect how the statistics in training set of an attribute influence the test performance.
>
> Q5: “it allows to study the impact of the number of occurrences in the training set. Is it more interesting than a learning curve ?”
> A5: The main contribution of this paper is not only to study how the occurrences of some attributes in training set influence different models, but also to investigate how different datasets are sensitive to occurrences of some attributes in the training set.  A learning curve is far from this goal.
>
> Q6: “multi attribute familiarity : risk of metric explosion ? how to select the attributes ?”
> A6: Multi-attribute familiarity may be a risk of metric explosion, but at the same time, it could
> encourage us to explore new meaningful measures. For example, “MF-et” could
> quantify the category ambiguity phenomenon. (Analogously, deep neural networks
> achieve impressive at the cost of architecture engineering.).  We would like to search for new combinations of attributes, which could be as our future work but is out of scope for this paper.
>
> Q7: “The encoding of the model name is not clear”
> A7: Although we have tried our best to name these 11 models more intuitive and more precise (by
> highlighted sub-words, detailed model choices), yet we still think we could do it better. We have added more explanation in our revised version.
>
> Q8: “a metric on all the dataset for each model could be computed to decide which one is the best
> overall”
> A8: We are not sure if we have understood your meaning of the term “metric”: Do you suggest we evaluate each model based on solely one attribute and find another best overall? Here, it would make no sense since different attributes just provide more fine-grained results, which will not lead to a new best overall performance.
>
> Q9: “analysis of Fig 4 : R-eLen does not existe (R-Ele). what is eta ?”
> A9: Thanks for catching the typos. We have corrected them: “R-Ele, R-eLen -> R-eLen”, eta -> zeta
>
> Q10: “figure 2 : where are the links to levels ?”
> A10: Fig.2 is used to aid the understanding of the “Attribute Definition” (Sec.3.1) and “Bucketization Strategy” (Sec.3.2).

---

> ### Author Response · Authors · 2019-11-14
> **Response to Review #1-Part3**
>
> For other detailed suggestions, we have refined our paper based on your feedback:
> 1)    Clarify the description of “Supplementary exam”
> 2)  Re-organize the Sec.2.2 and remove some repetition in methodological
> perspective
> 3)    Merge Sec.3 into Sec.2
> 4)    Add a more intuitive explanation of the measures defined in Sec.4.3 (3.3 in new version)
>
> Hope we address your concern correctly and look forward to your feedback again.

---

### Official Review · AnonReviewer2 · 2019-10-29
**Official Blind Review #2**

**Rating:** 8

**Review:**

This paper discusses a methodology to interpret models and model outputs for Named Entity Recognition (NER) based on assigned attributes. The key idea is to bucketize the test data based on characteristics of attributes and then comment on effect of the attribute on the model, the task itself or the dataset bias.

The empirical evaluation is impressive. The authors have constructed a series of experiments to make their case. The paper is well-written and easy to understand, albeit some of the related work seems a little unrelated to the task at hand. While the authors have tried to state that the method is "general" and goes beyond NER, I am not sure if that is the case. The creation of attribute buckets is vital for any further analysis, its not clear how the method can be adapted to more general settings unless such attributes and buckets can be created easily (e.g. using domain knowledge). Furthermore, there is only one problem setting considered (i.e. NER), and for the paper is make claim to more general settings, I would expect evaluations on atleast one more problem setting. I would suggest the authors modify the claims accordingly. This is not to diminish from their contributions in the NER.

The bucketization idea is not something out of the park novel. It is probably something already being used in practice. However, delineating the procedure and suggesting quantifiable statistics and designing experiments to illustrate how these can be used to draw qualitative conclusions is something that is very  interesting and useful to the community as a whole. The strongest part of this paper is the empirical evaluation that allows drawing interesting conclusions, and suggests a methodology to reach that conclusion. While some of the claims made (e.g. regarding dataset biases) probably require further and deeper analysis, this is a good first step that should foster further research and discussion.

**Experience Assessment:**

I have read many papers in this area.

**Review Assessment: Checking Correctness Of Derivations And Theory:**

I carefully checked the derivations and theory.

**Review Assessment: Checking Correctness Of Experiments:**

I assessed the sensibility of the experiments.

**Review Assessment: Thoroughness In Paper Reading:**

I read the paper at least twice and used my best judgement in assessing the paper.

---

> ### Author Response · Authors · 2019-11-14
> **Response to Review #2**
>
> Thank you for your encouraging review. We will continue to improve the draft in the revised version.
>
> Q1: “The bucketization idea is not something out of the park novel. It is probably something already being used in practice. However, delineating the procedure and suggesting quantifiable statistics and designing experiments to illustrate how these can be used to draw qualitative conclusions is something that is very interesting and useful to the community as a whole.”
> A1: We’re quite excited that you have pointed out the most challenging part of our work.
> Yes, when we would like to take multiple attributes, models, datasets all together, the most
> challenging thing is how to derive specific conclusions based on these tremendous results. In this paper, we overcome the difficulty by designing several meaningful measures, which can help us understand the relative merits between models quantitatively. This work also would like to show:
> when multiple datasets, models are ready, the time is ripe for us to shift the data-driven
> learning to data-driven analyzing (conduct an analysis over plenty of experimental data with the help of meaning measures)
>
> Q2: “While the authors have tried to state that the method is "general" and goes beyond NER, I am not sure if that is the case. The creation of attribute buckets is vital for any further analysis, its
> not clear how the method can be adapted to more general settings unless such
> attributes and buckets can be created easily (e.g. using domain knowledge).”
> A2: We try to address your concern by presenting a detailed description of general settings on other tasks. You could refer to our first answer to R3.
>
> Q3: “the paper is well-written and easy to understand, albeit some of the related work seems a little unrelated to the task at hand”
> A3: Thanks for your suggestion and we have made it revised in our new version.

---

### Author Response · Authors · 2019-11-14
**Updated version of the paper (Version 1.0)**

We thank all reviewers for their comments. They are extremely insightful and help us to make our paper better. We have been refining our paper based on their suggestions, and a new version is uploaded.
Below is a summary of the major changes:
1) We re-organize Section2,3 in the last version and merge them as "Preliminaries" Section to summarize the properties of evaluation methods (of related work) and describe the NER task and its current evaluation strategy. We remove some redundant description. (To address R2 and R3's concern)
2) We give more explanation of the "supplementary exam" to address R1's concern.
3) We refine Section3.2 to make the description of Fig.2 and Tab.2 more clear.
4) We make the introduction of R-Bucket more clear and give a concrete example.  Additionally, we have removed the last paragraph in Section4.2 (now it is Section3.2) (To address R3's concern)
5) We add an intuitive explanation for each measure defined in Section3.3. (To address R1's concern)
6) We add a more detailed explanation for the names of 11models and give an example. (To address R1's concern)
7) We refine our introduction section, providing detailed examples to show how to adapt to the proposed methodology to other types of NLP tasks.

---

### Decision · Program_Chairs · 2019-12-19

**Decision:**

Reject

**Comment:**

The paper diligently setup and conducted multiple experiments to validate their approach - bucketizating attributions of data and analyze them accordingly to discover deeper insights eg biases. However, reviewers pointed out that such bucketing is tailored to tasks where attributions are easily observed, such as the one of the focus in this paper -NER. While manuscript proposes this approach as ‘general’, reviewers failed to seem this point. Another reviewer recommended this manuscript to become a journal item rather than conference, due to the length of the page in appendix (17). There were some confusions around writings as well, pointed out by some reviewers. We highly recommend authors to carefully reflect on reviewers both pros and cons of the paper to improve the paper for your future submission.